# Impact of Perceived Stress, Locus of Control, and Self-Efficacy on Allergic Contact Dermatitis

**DOI:** 10.3390/healthcare13192498

**Published:** 2025-10-01

**Authors:** Francisco José Navarro-Triviño, Álvaro Prados-Carmona, Ricardo Ruiz-Villaverde, María Isabel Peralta-Ramírez

**Affiliations:** 1Department of Contact Eczema and Immunoallergic Diseases, Dermatology, University Hospital San Cecilio, 18007 Granada, Spain; 2Instituto Biosanitario de Granada, Ibs, 18016 Granada, Spain; apradoscar@gmail.com (Á.P.-C.); ismenios@gmail.com (R.R.-V.); 3Department of Dermatology, University Hospital San Cecilio, 18007 Granada, Spain; 4Escuela Internacional de Posgrado, University of Granada, 18016 Granada, Spain; 5Mind, Brain and Behavior Research Center (CIMCYC), University of Granada, 18016 Granada, Spain; mperalta@ugr.es; 6Department of Personality, Assessment and Psychological Treatment, Faculty of Psychology, University of Granada, 18016 Granada, Spain

**Keywords:** allergic contact dermatitis, perceived stress, locus of control, self-efficacy, psychosocial factors

## Abstract

**Background/Objectives**: Allergic contact dermatitis (ACD) is a chronic inflammatory disease with a high prevalence, affecting various aspects of patients’ lives. Psychosocial factors may influence disease management and outcomes, including perceived stress, locus of control, and self-efficacy. This study examines the presence of these factors in ACD and their association with disease severity and patient characteristics. **Methods**: A cross-sectional study included 225 adults with ACD and 225 healthy controls. Exclusion criteria were other skin diseases, psychiatric disorders, or intellectual disabilities. Sociodemographic and clinical variables, such as disease duration and severity, were recorded. Perceived stress, locus of control, and self-efficacy were assessed using validated questionnaires. Statistical analyses, including *t*-tests and multiple linear regression, were performed to explore group differences and predictors of clinical and psychosocial outcomes. **Results**: ACD patients exhibited higher perceived stress than controls (M = 39.36 vs. 24.74, *p* < 0.001), with stress levels correlating with disease severity (B = 0.062, 95% CI [0.050, 0.074], *p* < 0.001). Female sex (B = −5.896, *p* < 0.001) and lower education (B = −2.606, *p* = 0.035) predicted higher stress. Locus of control and self-efficacy showed statistically significant but modest differences between groups. **Conclusions**: Perceived stress was significantly associated with the severity of ACD, highlighting the necessity of incorporating psychological interventions into disease management. Programs focused on stress reduction and patient education should be integrated into clinical care to enhance outcomes. Longitudinal research is essential to establish causal relationships and evaluate the long-term benefits of tailored psychological support on disease progression and patient well-being.

## 1. Introduction

Allergic contact dermatitis (ACD) is a chronic inflammatory disease triggered by a type IV hypersensitivity reaction, primarily mediated by T lymphocytes after sensitization to allergens [1]. With a prevalence of 27% in Europe, ACD significantly impacts quality of life, particularly in women [2]. Symptoms range from acute erythema and vesicles to chronic lichenification and fissures, often causing physical discomfort and psychological distress [3].

Occupational ACD, particularly when affecting visible areas like the face and hands, is strongly associated with heightened levels of anxiety and depression [4]. Meta-analyses reveal that dermatitis increases the odds of mental health issues, including depression (OR 1.64) and anxiety (OR 1.68) [5], underscoring its substantial psychosocial burden.

Psychosocial factors, such as perceived stress, locus of control, and self-efficacy, are pivotal in chronic disease management. These are psychological constructs commonly used in the mental health field, and their application may be extrapolated to the context of ACD to better understand patients’ emotional responses and coping mechanisms. Psychological stress affects the skin through neuroendocrine–immune mechanisms, often described as part of the skin–brain axis. This bidirectional system integrates neural, endocrine, and immune pathways, providing a framework to understand how psychosocial factors may contribute to skin inflammation. Beyond cutaneous effects, accumulating evidence shows that chronic dermatoses such as atopic dermatitis can also predispose to mental health disorders and cognitive impairment through both psychosocial and biological mechanisms, including proinflammatory cytokine signaling across the blood–brain barrier and altered neurotransmitter pathways [6]. In this context, the skin–brain axis not only contributes to local inflammation but may also explain the higher risk of depression, suicidality, and other psychiatric comorbidities observed in patients with chronic inflammatory skin diseases. It can disrupt the epidermal barrier [7], activate the hypothalamic–pituitary–adrenal axis [8], and alter cytokine profiles [9], thereby promoting inflammation and impairing cutaneous homeostasis.

Self-efficacy is a psychological concept that refers to an individual’s belief in their own ability to successfully perform specific tasks or manage particular situations, whereas locus of control describes how individuals attribute the causes of events in their lives, reflecting beliefs about health outcomes. Both can influence treatment adherence. An internal locus of control refers to the belief that one’s actions influence outcomes, whereas an external locus of control reflects the perception that outcomes are determined. Internal locus correlates with better results [10]. Similarly, self-efficacy, or confidence in managing specific challenges, is pivotal for adherence to allergen avoidance strategies and disease management. Despite their importance, the role of these factors in ACD has not been thoroughly explored, highlighting the need for further research to understand their impact on disease management. Few studies have directly examined the interplay of these psychosocial factors in ACD, making this an important area for investigation.

Addressing this gap, the present study compares perceived stress, locus of control, and self-efficacy between ACD patients and healthy controls. In the ACD group, this study also examines the associations of these psychosocial factors with disease severity, duration, and frequency of emergency visits.

### Bulleted Summary

Perceived stress is significantly higher in patients with allergic contact dermatitis (ACD) compared to healthy controls, and it is positively associated with disease severity, emphasizing its clinical relevance.Sociodemographic factors, such as female sex and lower educational attainment, are significant predictors of higher stress levels in ACD patients, while locus of control and self-efficacy showed limited associations.The findings highlight the importance of integrating psychological support and stress management strategies into the comprehensive care of ACD patients to improve outcomes.

## 2. Materials and Methods

### 2.1. Patients

This cross-sectional study was conducted with patients referred to the Immunoallergy Unit of the Dermatology Department at a tertiary hospital in Spain between January 2021 and December 2023. Healthy controls were recruited primarily among companions or relatives of dermatology patients. This strategy was chosen for feasibility and to ensure comparability in key sociodemographic variables, including age, sex, and educational attainment. Both groups underwent dermatological evaluation to confirm eligibility. For controls, this ensured the absence of dermatological disease, psychiatric disorders, or psychopharmacological treatment.

The sample size was determined based on previous studies evaluating the impact of ACD. To minimize Type I and II errors, a significance level of 0.05 and a statistical power of 0.80 were used. A total of 225 ACD patients and 225 controls were recruited to ensure adequate statistical power.

Inclusion criteria for the ACD group included adults (≥18 years) with a confirmed diagnosis of ACD established through positive patch testing with clinically relevant allergens, with sufficient literacy in Spanish. In ACD patients, skin lesions were not restricted to the hands but could occur at different anatomical sites. Exclusion criteria comprised intellectual disabilities, diagnosed psychiatric disorders, current or previous psychopharmacological treatment, and other dermatological conditions unrelated to ACD, such as atopic dermatitis (AD), psoriasis, lichen planus, and drug eruption, among others.

The study was approved by the Institutional Medical Research Ethics Committee (DERM_HUSC_-2021) and conducted in accordance with the principles outlined in the Declaration of Helsinki. All participants received an information sheet detailing the study objectives, the confidential and anonymous handling of their data, and the voluntary nature of their participation. Written informed consent was obtained from all participants before their inclusion in the study.

### 2.2. Instruments

Sociodemographic and clinical variables were recorded in a structured medical interview. Disease duration was quantified in months based on the patient-reported time since symptom onset. Only sociodemographic data and mental health-related variables necessary to fulfill inclusion criteria were documented in the control group.

The diagnosis of ACD was confirmed using patch testing with the national standard series and extended/specific allergen series provided by Chemotechnique Diagnostics^®^ (Vellinge, Sweden), tailored to each case. Patch test results were interpreted according to the recommendations of the European Society of Contact Dermatitis (ESCD) [11]. Additionally, the clinical relevance of identified allergens was assessed, considering the source of exposure and the likelihood of causality.

Although physiological biomarkers, such as cortisol, were not included, this study focused on psychosocial self-reported measures due to their direct clinical relevance in ACD management.

### 2.3. Severity Variables

The medical Investigator Global Assessment (mIGA) scale [12] was used to assess ACD severity during the initial visit. This scale has demonstrated reliability indexes ranging from 0.516 to 0.778 and validity indexes between 0.497 and 0.893. The mIGA scale classifies severity levels as follows: 0 indicates the absence of lesions; 1 corresponds to very mild lesions with barely perceptible redness or scaling; 2 represents mild lesions with visible but limited redness or scaling; 3 reflects moderate involvement, characterized by clearly visible lesions with redness and scaling across multiple areas; and 4 indicates severe involvement, with extensive lesions showing intense redness, significant scaling and, in some cases, vesicles or erosions covering a large portion of the skin.

Patients’ self-perception of disease severity and concern about their lesions was evaluated using numerical rating scales (NRS) ranging from 0 to 10, where 0 represents no concern and 10 indicates the worst imaginable clinical condition.

### 2.4. Assessment of Stress, Self-Efficacy, and Locus of Control

Three standardized, validated, and Spanish-adapted questionnaires were used to evaluate stress, self-efficacy, and locus of control.

**Perceived Stress Scale (PSS)** [13]: Designed by Cohen et al., this tool is a validated and reliable measure, with Cronbach’s alpha values ranging from 0.78 to 0.91. It comprises 14 items that evaluate the individual’s subjective experience of stress over the preceding month. Responses are scored on a Likert scale from 0 (never) to 4 (very often), yielding a total score that can reach a maximum of 56. Higher scores reflect increased levels of perceived stress.

**Multidimensional Health Locus of Control (MHLC) Scale** [14]: This tool evaluates health-related locus of control across three dimensions: internal control, external control attributed to others (e.g., medical professionals), and external control attributed to chance or luck. It consists of 18 items divided into three subscales of 6 items each, rated on a scale from 1 (strongly disagree) to 6 (strongly agree), with reliability ranging from 0.70 to 0.87 (Cronbach’s alpha). Higher internal control scores reflect a belief in personal responsibility for health, promoting better self-management. Conversely, higher scores in external control by others or by chance suggest that health outcomes are perceived as dependent on external factors, often correlating with poorer disease management.

**General Self-Efficacy Scale (GSES)** [15]. This 10-item scale, developed by Baessler and Schwarzer, has a reliability of 0.80 (Cronbach’s alpha). Items are rated from 1 (strongly disagree) to 4 (strongly agree), with scores ranging from 10 to 40. Higher scores indicate greater confidence in managing challenging situations, supporting better disease management.

### 2.5. Statistical Analysis

Group differences in sociodemographic variables were examined using chi-square tests and Student’s *t*-tests. Differences in questionnaire scores between ACD and control participants were assessed with Student’s *t*-tests and two-way ANOVA, including sociodemographic variables as factors. Within the ACD group, additional analyses explored heterogeneity according to disease duration and emergency department (ED) visits. Given their skewed distribution, duration and ED visits were summarized as medians and interquartile ranges, and analyzed using Kruskal–Wallis and Spearman’s rho. Linear regression models were applied to identify predictors of perceived stress, self-efficacy, and locus of control. All assumptions underlying linear regression (normality of residuals, homoscedasticity, and absence of multicollinearity) were formally tested and adequately satisfied. A two-tailed *p* < 0.05 was considered significant. Analyses were conducted using SPSS v27.0 (IBM Corp., Armonk, NY, USA).

## 3. Results

### 3.1. Sample Description

A total of 450 participants were included in the study, comprising 225 patients diagnosed with ACD and 225 healthy individuals as the control group. The two groups were matched on key sociodemographic variables, with no statistically significant differences observed (Table 1).

In the ACD group, disease duration showed a highly skewed distribution, with a median of 21 months (IQR 8–72: range 1–600). Within the past 12 months, 38.2% of patients reported visiting the emergency department due to ACD. The number of visits was highly skewed, with a median of 0 (IQR = 0–0; range 0–12) and a mean of 0.72 (SD = 1.91).

The mean mIGA score was 2.31 (SD = 1.061). Regarding functional limitation, the mean NRS score was 6.29 (SD = 2.899), while the mean concern about lesions score was 7.18 (SD = 2.695). Self-perceived disease severity had a mean score of 5.37 (SD = 3.02).

### 3.2. Perceived Stress in ACD Patients vs. Control Group

Perceived stress, measured using the Perceived Stress Scale (PSS), was significantly higher in the ACD group compared to the control group. ACD patients had a mean score of 39.36 (SD = 11.03), while the control group had a mean score of 24.74 (SD = 9.28) (*p* < 0.001).

Regarding perceived stress levels, 2.2% of participants in the ACD group were classified as mild, 43.1% as moderate, 39.6% as high, and 15.1% as severe. In the control group, 19.7% reported mild stress, 31.4% moderate stress, 37.8% high stress, and 11.1% severe stress. Figure 1 illustrates the distribution of participants across stress categories, showing that ACD patients were predominantly classified as moderate, high, or severe, whereas controls were more often mild or moderate. Perceived stress, measured using the Perceived Stress Scale (PSS), was significantly higher in the ACD group compared to the control group. ACD patients had a mean score of 39.36 (SD = 11.03), while the control group had a mean score of 24.74 (SD = 9.28) (*p* < 0.001).

A two-way ANOVA was conducted to assess the effects of group (ACD vs. control), sociodemographic variables (age, sex, and marital status), and their interactions on total perceived stress scores. Group differences were significant (*p* < 0.001), with ACD patients reporting higher perceived stress levels than controls. Sex and marital status showed significant effects (*p* < 0.001), whereas age did not. No significant interactions were also found between group and sociodemographic variables, indicating that differences in perceived stress between groups were independent of these characteristics.

Within the ACD group, perceived stress scores were significantly higher in women (M = 33.67) than in men (M = 28.49) (*p* < 0.0001). Regarding marital status, single (M = 36.10) and widowed individuals (M = 38.14) had higher stress levels compared to married (M = 29.52) and divorced individuals (M = 28.38) (*p* < 0.0001). No significant differences in perceived stress were observed based on educational level or employment status in ACD patients. Pearson correlation analysis revealed a statistically significant negative correlation between age and perceived stress levels (r = −0.178, *p* = 0.007).

In the ACD group (*n* = 225), perceived stress scores did not significantly differ across disease duration tertiles (Short ≤12 months, Medium 13–49 months, Long >49 months) (Kruskal–Wallis *p* = 0.57). Similarly, Spearman’s rho revealed no significant correlation between continuous disease duration and perceived stress levels (ρ = −0.08, *p* = 0.39). Regarding emergency department (ED) visits, perceived stress did not vary significantly among categories (0, 1–2, ≥3 visits) (Kruskal–Wallis *p* = 0.71). The correlation between the number of ED visits and perceived stress was also not significant (ρ = −0.01, *p* = 0.93).

### 3.3. Locus of Control in ACD Patients vs. Control Group

Comparative analysis revealed that the ACD group scored significantly higher on the external locus of control subscale associated with chance or luck (M = 17.84, SD = 6.45) compared to the control group (M = 16.46, SD = 5.21) (*p* = 0.013). In contrast, no significant differences were observed in the internal or external locus of control attributed to others.

In the ACD group, correlation analysis identified a significant negative relationship between internal locus of control and educational level (r = −0.146, *p* = 0.028). Similarly, a significant negative correlation was found between the external locus of control attributed to others and educational level (r = −0.146, *p* = 0.028). No significant correlations were observed between locus of control subscales and other sociodemographic or clinical variables, including age, sex, marital status, or employment status.

When stratified by disease duration tertiles (Short ≤12 months, Medium 13–49 months, Long >49 months) and emergency department visit categories (0, 1–2, ≥3 visits), locus of control subscale scores did not differ significantly across groups (all Kruskal–Wallis *p* > 0.19). Consistently, Spearman’s rho analyses confirmed no significant correlations between either continuous disease duration or the number of ED visits and locus of control scores (all |ρ| ≤ 0.11, *p* > 0.23).

### 3.4. General Self-Efficacy in ACD Patients vs. Control Group

Comparative analysis revealed no significant differences in general self-efficacy scores between the ACD group (M = 76.15, SD = 15.43) and the control group (M = 77.89, SD = 14.50) (*p* = 0.614). Although differences in self-efficacy between groups were not statistically significant, their potential role in facilitating treatment adherence and stress management warrants further exploration.

Within the ACD group, self-efficacy was not significantly associated with sociodemographic or clinical variables such as age, sex, marital status, educational level, or disease duration. However, a clear relationship emerged with healthcare utilization: patients with lower self-efficacy reported more frequent emergency department visits. Specifically, those with ≥3 visits exhibited the lowest self-efficacy scores (median = 69 [IQR 57.5–77]), followed by those with 1–2 visits (median = 75.5 [IQR 66–83.8]), whereas patients without visits had the highest scores (median = 80 [IQR 65–87]) (Kruskal–Wallis *p* = 0.020). Consistently, the number of ED visits correlated negatively with self-efficacy (Spearman’s ρ = –0.24, *p* = 0.010), indicating that reduced self-efficacy may predispose patients to increased emergency healthcare utilization.

A summary of the results for perceived stress, locus of control, and self-efficacy in both groups is presented in Table 2.

### 3.5. Relationship Between Perceived Stress, Locus of Control, and Self-Efficacy with Disease Severity, Duration, and Emergency Visits

Multiple linear regression analysis using the “enter” method was performed to evaluate whether perceived stress, locus of control (internal and external), and self-efficacy predicted key clinical variables.

For ACD severity, the regression model was significant (*p* < 0.001), with perceived stress emerging as the only significant predictor (B = 0.062, 95% CI [0.050, 0.074], *p* < 0.001). This indicates that for every additional point in the perceived stress score, ACD severity significantly increased. No significant associations were observed between ACD severity and internal locus of control, external locus of control attributed to others, external locus of control attributed to chance, or self-efficacy.

None of the predictors demonstrated a significant relationship between disease duration and emergency visits.

## 4. Discussion

This study’s primary objective was to evaluate perceived stress, locus of control, and self-efficacy levels in patients with ACD compared to a control group. ACD patients reported significantly higher perceived stress scores, with most subjects exhibiting high or very high-stress levels. Significant differences were also observed in the external locus of control subscale associated with chance or luck.

Regarding sociodemographic characteristics, female participants were associated with higher perceived stress levels and greater scores in the external locus of control subscale (chance/luck) compared to males. Participants with lower educational levels were more inclined toward an external locus of control attributed to others. However, no significant associations were found between educational level and self-efficacy or internal locus of control.

From a clinical perspective, perceived stress was the only factor significantly of ACD severity. These findings highlight the need to implement educational strategies and psychological support aimed at strengthening self-efficacy, modifying locus of control, and reducing stress in patients with ACD.

Beyond sociodemographic influences, we also explored clinical markers of disease chronicity and severity. Disease duration did not show significant associations with perceived stress, locus of control, or self-efficacy, suggesting that the psychological burden of ACD is not solely explained by chronicity. In contrast, the number of emergency department visits emerged as a significant indicator: patients with recurrent visits reported lower self-efficacy levels, and a negative correlation was observed between ED visits and self-efficacy. These findings suggest that acute disease exacerbations requiring urgent care may undermine patients’ confidence in their ability to manage ACD, highlighting the importance of targeted interventions to reinforce self-efficacy in those with frequent healthcare utilization.

Stress is a well-recognized aggravating factor in multiple dermatological conditions. Moreover, its impact is associated with increased disease severity [16], poorer quality of life, impaired sleep, and a higher risk of depression [17]. The findings of our study suggest that ACD patients exhibit significantly elevated levels of perceived stress, which may impose a substantial emotional and psychological burden. Both acute and chronic stress are known to exacerbate various skin conditions [18]. In particular, chronic stress can induce neuroendocrine and neuroimmune responses that contribute to persistent skin inflammation, thereby increasing lesion severity.

Our observations align with previous studies on chronic hand eczema, where 67.7% of patients reported high stress levels, regardless of etiological or morphological subtype [19]. Additionally, stress has been identified as a trigger for acne lesions, flares, and worsening of conditions such as AD, psoriasis, and hidradenitis suppurativa (HS) [20].

Notably, our results indicate that ACD patients experience higher perceived stress levels, as is the case also in other dermatoses, such as acne and psoriasis [20]. Consistent with other studies, we highlight the significant correlation between higher perceived stress and disease severity in patients with ACD, psoriasis, AD, and acne. In contrast, perceived stress appears to be independent of disease severity in HS patients. In a recent study on chronic urticaria patients, a significant negative correlation was found between perceived stress levels and quality of life [21].

These findings have significant implications. Beyond the emotional consequences associated with high levels of stress, it is crucial to consider stress’s ability to disrupt the neuroendocrine–immune axis, thereby perpetuating inflammation [22]. This pathophysiological basis reinforces the concept that dermatoses accompanied by pruritus are often associated with higher perceived stress levels, potentially creating a vicious cycle [23].

Psychological support provided to dermatology patients is estimated to be below 15%, according to published studies [20]. This underscores the need to integrate specialized professionals into dermatology to effectively address these patients’ psychological needs.

Documented pathophysiological mechanisms support our findings regarding the high levels of perceived stress in ACD patients. While this study did not evaluate physiological biomarkers, previous research has highlighted the role of salivary cortisol as a measure of psychological stress in chronic hand eczema [24]. However, its effectiveness in chronic stress scenarios, such as AD [25], is limited due to compensatory decreases in cortisol levels. More stable cortisol sources, such as follicular units, have been proposed for future research.

Additionally, psychological stress has been shown to impair epidermal barrier integrity. A recent study demonstrated a positive correlation between stress levels and transepidermal water loss measured by the Tewameter^®^ TM300, further emphasizing the impact of stress on skin function [26]. Additionally, research has demonstrated that psychological stress negatively affects epidermal barrier permeability, leading to increased severity and prolonged duration of inflammatory dermatoses [27]. These findings align with the observed relationship between perceived stress and disease severity, emphasizing the need to investigate these physiological pathways in future studies.

The type of locus of control can influence adherence to treatment in ACD patients. An internal locus of control facilitates compliance with medical recommendations [28] and fosters a greater commitment to self-care [29]. Conversely, an external locus of control, where health outcomes are attributed to chance or external factors, may lead to demotivation and lower treatment adherence. Our findings underscore the need for educational and patient-empowerment interventions to strengthen an internal locus of control to improve treatment adherence.

Although no significant differences in self-efficacy were found between the two groups, we would like to emphasize the potential role of self-efficacy in managing ACD. Self-efficacy directly influences treatment adherence and stress management.

Based on our findings, we hypothesize that patients with high self-efficacy may be more diligent at avoiding triggers, such as allergens identified through patch testing. Self-efficacy likely promotes favorable outcomes in ACD patients by enhancing their ability to adopt necessary lifestyle changes. Additionally, patients with higher self-efficacy are better equipped to manage stress and can break the itch–scratch cycle more effectively. Improved disease management, in turn, reduces the psychosocial burden associated with ACD [30].

Importantly, our findings also showed that self-efficacy plays a key role in healthcare utilization. Patients with higher self-efficacy appeared more capable of managing disease flare-ups and therefore relied less on emergency services. In contrast, individuals with lower self-efficacy reported more frequent emergency visits, suggesting that limited confidence in self-management may lead to earlier recourse to urgent care when facing exacerbation. Given that self-efficacy is a relatively stable psychological trait, these results underscore its importance as a protective factor against the overuse of emergency services. Strengthening self-efficacy in ACD could therefore contribute not only to better disease control but also to reducing healthcare burden and improving patient well-being. These findings emphasize the critical need to integrate clinical and psychological approaches in the comprehensive management of ACD patients, aiming to address the disease’s physiological and psychosocial aspects effectively. Incorporating psychological and educational interventions into clinical practice is crucial to addressing perceived stress, strengthening self-efficacy, and improving treatment adherence. Strategies such as self-help programs, cognitive-behavioral therapy (CBT), and psychoeducational approaches have demonstrated effectiveness in reducing stress and anxiety, particularly in patients experiencing the itch–scratch cycle [31]. Our findings emphasize the need for early identification of at-risk patients, such as women, single, or widowed individuals, to provide timely and targeted psychological support. Additionally, professions with high emotional demands, such as physicians and dentists, require specific strategies to manage occupational stress and prevent skin health deterioration [32]. Promoting an internal locus of control through coping and problem-solving programs is essential to empower patients and optimize disease management [33]. Addressing stress from a comprehensive and continuous perspective may help reduce disease severity and improve subjective measures of well-being, ensuring a more complete approach to health in ACD patients [33].

Interventions such as self-help programs, psychological therapy, and psychoeducational initiatives have shown potential in addressing and mitigating the impact of stress, particularly in patients with ACD. Educational strategies, including self-care, trigger management, and cognitive-behavioral therapy (CBT) for stress control, enhance self-efficacy and reduce anxiety, thereby improving treatment adherence [34]. Promoting an internal locus of control through specific coping and problem-solving programs is essential to optimizing ACD management and improving patient well-being. Future studies should employ longitudinal designs to confirm the causal relationships between perceived stress and clinical outcomes in ACD.

The main limitations of this study are its cross-sectional design and its conduction at a single center, which restrict causal inference and generalizability. Recruiting controls among patients’ companions may have introduced selection bias, although strict exclusion criteria were applied to minimize this risk. In addition, neither longitudinal follow-up nor validated itch severity measures were included, and no bioclinical or biomolecular parameters were collected, as these were beyond the scope of the study. Future research integrating biomarkers and longitudinal designs will be necessary to confirm and extend our findings.

## 5. Conclusions

This study highlights the strong association between perceived stress and ACD severity, underscoring its role as the primary psychosocial factor to be addressed in clinical management. By contrast, locus of control and self-efficacy showed only modest or non-significant associations, suggesting a secondary influence. Given the cross-sectional design, causality cannot be inferred. Future paraclinical and longitudinal studies will be essential to clarify causal pathways and to further investigate the integration of psychosocial and biological determinants in ACD.

## Figures and Tables

**Figure 1 healthcare-13-02498-f001:**
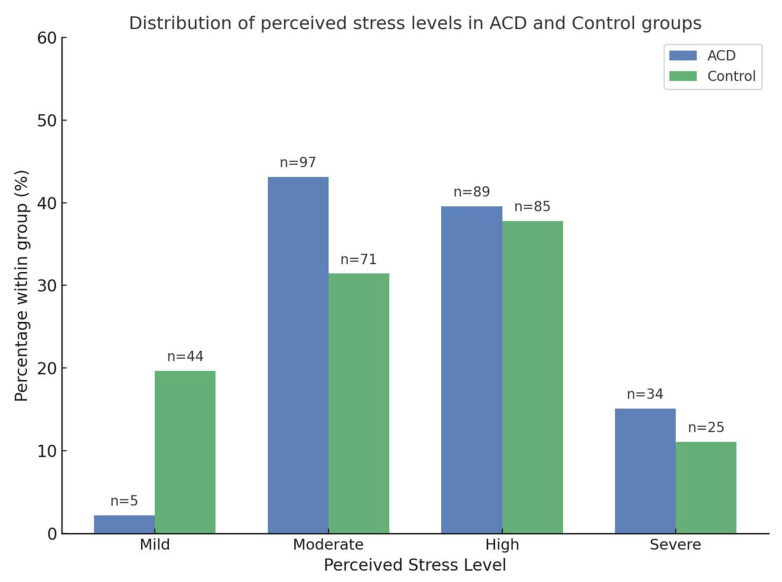
Distribution of perceived stress levels in ACD and control groups. Bars represent the percentage of participants within each group classified as Mild (0–10), Moderate (11–21), High (22–40), or Severe (41–56) according to the 14-item Perceived Stress Scale (PSS-14). Absolute numbers of participants (*n*) are indicated above each bar. The figure was created with assistance from ChatGPT-4 (OpenAI, 2024).

**Table 1 healthcare-13-02498-t001:** Sociodemographic Characteristics of Participants in the ACD and Control Groups.

Sociodemographic Variables	ACD Group (*n* = 225) M(SD)/*n* (%)	Control Group (*n* = 225) M(SD)/*n* (%)	*p*-Value (t/Chi)
**Age**	33.01 (4.561)	33.56 (4.327)	0.468
**Sex**			0.611
Women	152 (67.6%)	157 (69.8%)	
Men	73 (32.4%)	68 (30.2%)	
**Marital Status**			0.953
Single	76 (33.8%)	80 (35.6%)	
Married	117 (52%)	111 (49.3%)	
Divorced	23 (10.2%)	24 (10.7%)	
Widowed	9 (4%)	10 (4.4%)	
**Educational Level**			0.592
Basic Education	16 (7.1%)	18 (7.8%)	
Primary Education	19 (8.4%)	19 (8.4%)	
Secondary Education	38 (16.9%)	32 (14.2%)	
High School	39 (17.3%)	41 (18.2%)	
University Studies	113 (50.2%)	115 (51.1%)	
**Employed**	138 (61.33%)	124 (55.11%)	0.214

**Table 2 healthcare-13-02498-t002:** Comparison of perceived stress, locus of control, and self-efficacy between ACD patients and healthy controls. Values are presented as mean (SD). *p*-values were obtained using independent-sample *t*-tests.

Variable	ACD Group (*n* = 225) Mean (SD)	Control Group (*n* = 225) Mean (SD)	*p*-Value
Perceived Stress (PSS, total)	39.36 (11.03)	2474 (9.28)	<0.001
Internal Locus of Control	22.10 (5.12)	21.68 (4.98)	0.370
External Locus of Control (Others)	21.41 (5.75)	20.71 (5.23)	0.176
External Locus of Control (Chance)	17.84 (6.45)	16.46 (5.21)	0.013
Total Locus of Control	61.36 (13.80)	58.84 (11.67)	0.038
Self-Efficacy (GSES, total)	76.15 (15.43)	77.89 (14.50)	0.218

## Data Availability

The data supporting the findings are available from the corresponding author, FJNT, upon reasonable request (due to privacy restrictions and ethical considerations, as they contain sensitive clinical information from patients; this limitation is in accordance with data protection regulations and the recommendations of the ethics committee that approved the study). All the authors had full access to all the data in this manuscript and jointly assumed responsibility for its integrity.

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
