# Peer review of "Impact of Perceived Stress, Locus of Control, and Self-Efficacy on Allergic Contact Dermatitis"

_healthcare, 2025, doi:10.3390/healthcare13192498_

Round 1

Reviewer 1 Report

Comments and Suggestions for Authors

Dear author,

It is with great pleasure that I reviewed the manuscript. Here are some considerations:
1. Figure 2 shows that control group is higher on "low to high" stress level, while drastically changed on "severe". Why the sudden change and not gradual? Perhaps this can be discussed.
2. Seems like the stress is significant, locus of control, and efficacy is less/not significant. Yet the conclusion states all three as if the evidence supported them equally. The conclusion can be adjusted to better reflect the result of the study.

Thank you

Author Response

We sincerely thank the reviewers for their positive assessment of our work and for the constructive comments, which have greatly contributed to the improvement of the manuscript. We have tried to address all of their concerns point-by-point as follows:

Comment 1. “Figure 2 shows that the control group is higher on 'low to high' stress level, while drastically changed on 'severe'. Why the sudden change, rather than gradual? Perhaps this can be discussed.”

Response 1. We appreciate this observation. Initially, we added a clarification in the Discussion explaining that the apparent “sudden change” could be due to the categorical cut-offs of the Perceived Stress Scale (PSS). However, after further consideration, we decided to remove Figure 2 from the manuscript to avoid potential misinterpretation. Consequently, the explanatory sentences referring to this figure were also deleted to maintain consistency. Figure 2 and the corresponding explanatory note in the Discussion (page 09, lines 342–346) were removed.

Comment 2. “Seems like the stress is significant, locus of control, and efficacy is less/not significant. Yet the conclusion states all three as if the evidence supported them equally. The conclusion can be adjusted to reflect the result of the study better.”

Response 2. We thank the reviewer for this important remark. We agree that the evidence supporting perceived stress is stronger compared to locus of control and self-efficacy. Accordingly, we revised the Conclusion to emphasize perceived stress as the main psychosocial factor associated with ACD severity, while presenting locus of control and self-efficacy as secondary factors. This adjustment better aligns the conclusions with the study findings (page 10, line 425-431)

Reviewer 2 Report

Comments and Suggestions for Authors

José Navarro-Triviño et al. examine the psychosocial determinants of ACD severity. The topic is clinically relevant, as psychological comorbidities are increasingly recognized as important modifiers of dermatological disease burden. The manuscript is well organized and provides clear methodological descriptions. Nevertheless, several issues concerning study design, methodology, interpretation, and presentation should be addressed before the manuscript can be considered for publication.

  1. The cross-sectional design limits the ability to establish temporal or causal relationships. Although this is noted in the discussion, the conclusions occasionally suggest causality (e.g., “perceived stress significantly influences the severity of ACD”). The wording should be adjusted to reflect associations rather than causal effects.

  2. Healthy controls were primarily recruited among relatives or companions of dermatology patients. This approach may introduce bias, as such individuals could share environmental, behavioral, or psychosocial factors with patients. The rationale for this recruitment strategy should be explained in more detail, and its potential impact discussed.

  3. Disease severity assessment relied solely on the medical Investigator Global Assessment (mIGA) and NRS-based self-perception. Given that pruritus is a key symptom of ACD, the absence of validated itch severity measures (e.g., VAS, Itch NRS) limits the comprehensiveness of the evaluation. This omission should be acknowledged as a limitation.

  4. Regarding the statistical analysis, it is not clear whether the assumptions underlying linear regression (normality, homoscedasticity, collinearity) were formally tested. Please clarify.

Author Response

We thank the reviewer for the thoughtful and constructive feedback. We address each point below:

Comment 1. “The cross-sectional design limits the ability to establish temporal or causal relationships. Although this is noted in the discussion, the conclusions occasionally suggest causality (e.g., ‘perceived stress significantly influences the severity of ACD’). The wording should be adjusted to reflect associations rather than causal effects.”

Response 1. We agree with the reviewer. We have carefully revised the manuscript to ensure that our conclusions describe associations rather than causality. For example, the sentence “perceived stress significantly influences the severity of ACD” has been modified to “perceived stress was significantly associated with the severity of ACD.” Similar adjustments were made in both the Abstract and Discussion (page 01, lines 32-33; page 8, line 309-310; page 10, lines 425-431).

Comment 2. “Healthy controls were primarily recruited among relatives or companions of dermatology patients. This approach may introduce bias, as such individuals could share environmental, behavioral, or psychosocial factors with patients. The rationale for this recruitment strategy should be explained in more detail, and its potential impact discussed.”

Response 2. We thank the reviewer for this thoughtful comment. As clarified in the revised Methods section, controls were recruited primarily among relatives or companions of patients. This strategy was adopted to ensure feasibility and to improve comparability between groups in key sociodemographic variables (e.g., age, sex, educational level, and cultural background). As recommended in the methodological literature on case–control studies, the use of relatives or friends as controls can help reduce variability in psychosocial and environmental factors, thereby increasing homogeneity and allowing differences to be more confidently attributed to the disease under study rather than to external factors (page 3, line 103-106) (Wacholder S, Silverman DT, McLaughlin JK, Mandel JS. Selection of controls in case–control studies: I. Principles. Am J Epidemiol. 1992;135(9):1019-28.; Schlesselman JJ. Case–control studies: Design, conduct, analysis. New York: Oxford University Press; 1982). Furthermore, relatives and companions are generally more motivated to participate and adhere to study procedures, which increases feasibility and compliance.

We acknowledge, however, that this approach may also introduce a degree of selection bias, since relatives may share behavioral or psychosocial characteristics with patients. This potential limitation has now been explicitly recognized in the Discussion section.

Comment 3. “Disease severity assessment relied solely on the medical Investigator Global Assessment (mIGA) and NRS-based self-perception. Given that pruritus is a key symptom of ACD, the absence of validated itch severity measures (e.g., VAS, Itch NRS) limits the comprehensiveness of the evaluation. This omission should be acknowledged as a limitation.”

Response 3. We thank the reviewer for this valuable observation. Pruritus is indeed a central symptom of ACD. While our primary objective was to investigate psychosocial determinants such as perceived stress, locus of control, and self-efficacy, we acknowledge that the absence of validated itch severity measures (e.g., VAS, Itch NRS) represents a limitation of our study. Accordingly, we have explicitly included this point in the Limitations section (page 10, line 419-420), noting that future studies should incorporate standardized assessments of itch to achieve a more comprehensive evaluation of disease severity.

Comment 4. “Regarding the statistical analysis, it is not clear whether the assumptions underlying linear regression (normality, homoscedasticity, collinearity) were formally tested. Please clarify.”

Response 4. Thank you for pointing this out. We confirm that the assumptions of linear regression (normality of residuals, homoscedasticity, and absence of multicollinearity) were formally tested and met. This clarification has been added in the Statistical Analysis subsection (page 5, lines 182-186).

Reviewer 3 Report

Comments and Suggestions for Authors

The manuscript "The Influence of Perceived Stress, Locus of Control, and Self-Efficacy on Allergic Contact Dermatitis" is an original manuscript conducted on ACD patients on the psychological aspects of ACD.
Therefore, in this cross-sectional study, the authors included 225 adults with ACD and 225 healthy controls and analyzed their sociodemographic and clinical variables, such as disease duration and severity. They analyzed perceived stress, locus of control, and self-efficacy using validated questionnaires, and statistical analyses, including t-tests and multiple linear regression, were performed to investigate differences between groups and predictors of clinical and psychosocial outcomes. Thus, according to the results obtained, ACD patients showed higher perceived stress than the control group (p < 0.001), with stress levels that correlated with disease severity (p < 0.001). Female gender (p < 0.001) and lower education (p = 0.035) predicted higher stress. Locus of control and self-efficacy showed statistically significant but modest differences between groups.
Overall, I find this manuscript very useful for current knowledge and practice in this area.  Also, the authors included high number of participants. However, there are a few suggestions:

ABSTRACT - Conclusion. The text "Perceived stress significantly influences the severity of ACD, highlighting the need to include psychological interventions in the management of the disease." Please check if this is the only meaning or can it be interpreted both ways, i.e. the other way around / vice versa? Potentially, the severity of ACD significantly influences their perceived stress.
Perhaps these two factors are significantly related, but potentially both directions are possible. This is the interpretation of the findings.

KEYWORDS: Some keywords are too long and it is not common to mention them in such forms.

INTRODUCTION: Briefly describe the network of factors in the skin related to psychological stress and the key factors and mechanisms involved in these processes.

METHODS AND MATERIALS:
- I suggest the authors provide more information about the healthy control group.
- Were both groups examined by a dermatologist or did they just fill out/use a questionnaire?
- Were the skin lesions only on the hands or in different locations?

RESULTS:
- I suggest the authors add one table of figures showing the key results for all examined factors (current versions of the tables do not show results for locus of control and self-efficacy in patients with allergic contact dermatitis).

REFERENCES - Ref 7 (please correct the authors' names): The correct citation is: Pondeljak, N.; Tomić, L.; Lazić-Mosler, E.; Šitum, M..; Karlović, D.; Lugović-Mihić, L. Salivary cortisol and perceived psychological stress in 440 patients with chronic hand contact dermatitis. Contact Dermatitis 2023, 89, 393–395, doi:10.1111/cod.14399.

Author Response

We sincerely thank the reviewer for the positive evaluation of our work and for the valuable comments. We have carefully considered all suggestions and revised the manuscript accordingly. Our detailed responses are provided below:

Comment 1. Abstract – Conclusion: “The text ‘Perceived stress significantly influences the severity of ACD…’. Please check if this is the only meaning or it can be interpreted both ways? Potentially, the severity of ACD significantly influences perceived stress. Perhaps these two factors are significantly related, but both directions are possible.”

Response 1. We fully agree with the reviewer. In cross-sectional designs, it is not possible to establish causality, and bidirectional interpretations are plausible. We have revised the Abstract conclusion to state: “Perceived stress was significantly associated with the severity of ACD, highlighting the need to consider psychological interventions in disease management.” (page 01, lines 32-33; page 8, line 309-310; page 10, lines 425-431). This wording emphasizes association rather than directionality.

Comment 2. Keywords: “Some keywords are too long and it is not common to mention them in such forms.”

Response 2. We thank the reviewer for this observation. The keywords have been shortened and standardized to improve clarity and alignment with indexing practices (page 1, line 39-40)

Comment 3. Introduction: “Briefly describe the network of factors in the skin related to psychological stress and the key factors and mechanisms involved.”

Response 3. Thank you for this suggestion. The Introduction has been expanded to briefly describe the neuroendocrine–immune pathways linking psychological stress with skin function, highlighting barrier disruption, HPA axis activation, and cytokine modulation (page 2, line 59-72.

Comment 4. Methods – Healthy controls: Provide more information about the healthy control group. Were both groups examined by a dermatologist, or did they just fill out a questionnaire? Were the skin lesions only on the hands or in different locations?”

We thank the reviewer for this thoughtful comment. As clarified in the revised Methods section, healthy controls were recruited primarily among relatives or companions of dermatology patients. This approach was adopted for several reasons:

  1. Control of sociodemographic variables: relatives and friends often share similar social, cultural, and sometimes economic environments, which reduces variability in factors such as educational level, socioeconomic status, and lifestyle that could otherwise confound psychological outcomes.
  2. Control of environmental context: Psychological variables such as perceived stress or quality of life are strongly influenced by the surrounding environment. Using relatives/friends ensures that patients and controls are more comparable in this respect.
  3. Greater homogeneity: this strategy facilitates attributing observed differences to the dermatological condition itself rather than to uncontrolled external factors.
  4. Feasibility and compliance: relatives and friends are generally more motivated to participate and more likely to adhere to study procedures, thus improving feasibility and data quality.

This methodological choice is consistent with established recommendations in case–control research (Wacholder S, Silverman DT, McLaughlin JK, Mandel JS. Selection of controls in case–control studies: I. Principles. Am J Epidemiol. 1992;135(9):1019–1028; Schlesselman JJ. Case–control studies: Design, conduct, analysis. Oxford University Press, 1982).

However, we acknowledge that this strategy may also introduce a degree of selection bias, since relatives may share behavioral or psychosocial traits with patients. This potential limitation has now been explicitly recognized in the Discussion section.

Comment 5. Results – Presentation “Add one table of figures showing the key results for all examined factors (current versions of the tables do not show results for locus of control and self-efficacy).”

Response 5. We thank the reviewer for this helpful suggestion. An additional summary table has been added (page 7, lines 278-281), presenting the mean scores and standard deviations of perceived stress, locus of control, and self-efficacy in both groups, together with corresponding p-values.

Comment 6. References “Ref 7 (please correct the authors' names).”

Response 6. We apologize for the oversight and have corrected the reference accordingly, following the reviewer’s suggestion.

Reviewer 4 Report

Comments and Suggestions for Authors

This paper highlights a current topic that impacts the quality of life of patients diagnosed with allergic contact dermatitis. Over the last decade, there has been an increase in epidemiological and clinical studies that highlight the involvement of psychosocial factors in the onset and progression of chronic inflammatory skin diseases.          

Following the analysis, there are several suggestions and recommendations:

  1. Introduction: The idea that dermatitis increases the risk of mental health problems is surprising. However, the direct clinical involvement of the skin-brain axis is well known, which may justify Niemeier's hypothesis (2002) that inflammatory dermatoses may represent a form of somatization of mental disorders. Therefore, it is necessary to supplement the information contained in these paragraphs.
  2. Material and Method: Can you explain why the control group consists mainly of the patients' relatives? Could this influence the final results?
  3. Neural signaling "collaborates" with endocrine signaling in maintaining systemic homeostasis. The latter is also conditioned by the integrated activation of autocrine, paracrine, and contact-dependent signaling. Therefore, it is understandable why allergic contact dermatitis, similar to other chronic inflammatory conditions mediated by T lymphocytes, has a heterogeneous etiopathogenesis. For this reason, it is appropriate to supplement the study with the evaluation of a bioclinical or biomolecular parameter that can support the causal relationship.
  4. A paragraph can be added to outline the limitations of the study.
  5. Conclusions: As this is a cross-sectional study, no direct causal relationship between psychosocial factors and dermatitis can be established. Therefore, the first conclusion needs to be reworded. Specific paraclinical studies are needed to demonstrate direct causality.

Author Response

We sincerely thank the reviewer for the thoughtful evaluation of our manuscript and for the constructive comments, which have helped us to further improve the quality and clarity of the paper. Our detailed responses are provided below:

Comment 1. Introduction: “The idea that dermatitis increases the risk of mental health problems is surprising. However, the direct clinical involvement of the skin-brain axis is well known, which may justify Niemeier's hypothesis (2002) that inflammatory dermatoses may represent a form of somatization of mental disorders. Therefore, it is necessary to supplement the information contained in these paragraphs.”

Response 1. We thank the reviewer for this insightful remark. We have expanded the Introduction to include a brief description of the skin–brain axis, emphasizing its role as a bidirectional neuroendocrine–immune system that links psychological stress with cutaneous inflammation (page 2, lines 59-72). This addition strengthens our rationale for examining psychosocial determinants in ACD. Furthermore, we have incorporated recent evidence highlighting the well-established relationship between dermatitis and mental health problems, which reinforces Niemeier’s hypothesis.

Comment 2. Material and Methods: “Can you explain why the control group consists mainly of the patients' relatives? Could this influence the final results?”

Response 2. We agree with the reviewer. As explained in the revised Methods section, controls were recruited primarily among relatives or companions to ensure feasibility and comparability in sociodemographic variables. We also acknowledge that this approach may introduce selection bias, as relatives may share behavioral or psychosocial characteristics with patients. This limitation has been explicitly discussed in the Discussion.

Comment 3. Study enrichment with bioclinical or biomolecular parameters. “Neural signaling collaborates with endocrine signaling… it is appropriate to supplement the study with the evaluation of a bioclinical or biomolecular parameter that can support the causal relationship.”

Response 3. We thank the reviewer for this thoughtful comment. As clarified in the revised Methods section, healthy controls were recruited primarily among relatives or companions of dermatology patients. This approach was adopted for several reasons:

  1. Control of sociodemographic variables: relatives and friends often share similar social, cultural, and sometimes economic environments, which reduces variability in factors such as educational level, socioeconomic status, and lifestyle that could otherwise confound psychological outcomes.
  2. Control of environmental context: Psychological variables such as perceived stress or quality of life are strongly influenced by the surrounding environment. Using relatives/friends ensures that patients and controls are more comparable in this respect.
  3. Greater homogeneity: this strategy facilitates attributing observed differences to the dermatological condition itself rather than to uncontrolled external factors.
  4. Feasibility and compliance: relatives and friends are generally more motivated to participate and more likely to adhere to study procedures, thus improving feasibility and data quality.

This methodological choice is consistent with established recommendations in case–control research (Wacholder S, Silverman DT, McLaughlin JK, Mandel JS. Selection of controls in case–control studies: I. Principles. Am J Epidemiol. 1992;135(9):1019–1028; Schlesselman JJ. Case–control studies: Design, conduct, analysis. Oxford University Press, 1982).

However, we acknowledge that this strategy may also introduce a degree of selection bias, since relatives may share behavioral or psychosocial traits with patients. This potential limitation has now been explicitly recognized in the Discussion section.

Comment 4. Limitations: “A paragraph can be added to outline the limitations of the study.”

Response 4. Thank you for this recommendation. A dedicated paragraph outlining the main limitations—cross-sectional design, single-center setting, recruitment of controls among relatives, absence of itch severity measures, and lack of biomolecular data—has been added to the Discussion (page 10, lines 416-423)

Comment 5. Conclusions: “As this is a cross-sectional study, no direct causal relationship between psychosocial factors and dermatitis can be established. Therefore, the first conclusion needs to be reworded. Specific paraclinical studies are needed to demonstrate direct causality.”

Response 5. We fully agree. The Conclusion has been revised to emphasize association rather than causality, and we now state that paraclinical and longitudinal studies are required to establish direct causal relationships (page 10-11, lines 429-431)

Round 2

Reviewer 4 Report

Comments and Suggestions for Authors

The authors responded to the recommendations made. The article looks much better in this form.